## [Decision Letter · Decision Letter 0]

25 Aug 2019

PONE-D-19-15923

Incidence, trends, and outcomes of infection sites among patients with sepsis: a nationwide study

PLOS ONE

Dear Dr Lee,

Thank you for submitting your manuscript to PLOS ONE. After careful consideration, we feel that it has merit but does not fully meet PLOS ONE’s publication criteria as it currently stands. Therefore, we invite you to submit a revised version of the manuscript that addresses the points raised during the review process.

The reviewers raised some concerns regarding the chosen sepsis definition, particularly as up to 30% of patients with sepsis do not have positive cultures and therefore would not be captured in the current analysis. Therefore, sensitivity analysis using different sepsis definition criteria (implicit Angus criteria vs. explicit sepsis criteria) should be performed to corroborate the presented results. In addition, ICD-9 codes used to define 'organ dysfunction' should be listed in the supplement. Second, it is unclear whether the analyses included both community-acquired sepsis ('present on admission') or nosocomial sepsis. These are very different entities with regards to epidemiology, source of infection, risk of death, etc. Along these lines it would be interesting to know what proportion of hospitalizations were primary 'medical' or 'surgical'. Third, previous studies by Lindenauer (JAMA 2012) and Rhee (JAMA 2017) suggest that temporal trends in pneumonia and sepsis estimates are associated with differences in coding - additional sensitivity analysis should be performed to verify or refute these findings. An additional limitation is the inability of administrative data to establish a firm temporal relationship between sepsis and the onset of organ dysfunction. This important limitation should be added to the discussion section. Finally, the discussion should be expanded to contrast the findings to preexisting literature. For example,  Zahar JR et al previously reported no association between infection site and presence of bacteremia with mortality. 

We would appreciate receiving your revised manuscript by Oct 09 2019 11:59PM. To enhance the reproducibility of your results, we recommend that if applicable you deposit your laboratory protocols in protocols.io, where a protocol can be assigned its own identifier (DOI) such that it can be cited independently in the future. For instructions see: http://journals.plos.org/plosone/s/submission-guidelines#loc-laboratory-protocols

We look forward to receiving your revised manuscript.

Kind regards,

Florian B. Mayr

Academic Editor

PLOS ONE

Journal Requirements:

2. Please include your tables as part of your main manuscript and remove the individual files. Please note that supplementary tables (should remain/ be uploaded) as separate "supporting information" files

This work is partly supported by NTUH.107-P03 grant

Reviewers' comments:

Reviewer's Responses to Questions

**Comments to the Author**

1. Is the manuscript technically sound, and do the data support the conclusions?

Reviewer #1: Partly

2. Has the statistical analysis been performed appropriately and rigorously? 

Reviewer #1: Yes

3. Have the authors made all data underlying the findings in their manuscript fully available?

Reviewer #1: Yes

4. Is the manuscript presented in an intelligible fashion and written in standard English?

Reviewer #1: Yes

5. Review Comments to the Author

Reviewer #1: Mann et al. have conducted a longitudinal analysis of sepsis outcomes by site of infection, over an extended period, using the National Inpatient Sample database, representing a large portion of the U.S. population. The authors have performed an appropriate analysis based on the data available and provided trends in mortality by infectious site over time, an important finding. However, I have some reservations about the methodology used provided in comments below.

Major revisions:

1. Methods: The authors’ primary conclusion in this study is that site of infection affects mortality and is changing over time. However, identification of site of infection seems inconsistent.

a. (Supplemental Table 2) Certain infections appear to be omitted from the inclusion diagnoses (i.e. cholangitis, Clostridium dificile colitis, CNS infections, endocarditis, and GU infections in women such as endometritis or ovarian abscess). Suggest improving identification of source of infection as this is the most important variable in the analysis.

b. Second, how do the authors identify the primary site of infection? For example, if a hospitalization is coded for intestinal perforation and bacteremia and fungemia, how would this hospitalization be grouped in their analysis? Would it be the highest ranking diagnosis code or would it be included in all groups. This is important for assumptions of mortality in each group. It appears that these groups are mutually exclusive but it is not clear in the methods section.

c. Does primary bacteremia mean no other diagnosis could be found for site of infection? More detail is required since there is a significant decrease in this site of infection over time. I suspect the outcomes and numbers of this site of infection would change greatly once item 1a is addressed.

2. Methods Line 95. The authors identify sepsis using methods by Martin et al. There are several other methods for identifying sepsis including Dombrovskiy et al. (PMID: 17414736). To improve the confidence of the conclusions, a sensitivity analysis using one of the other methods would be helpful.

3. Methods Line 126. The adjusted analysis includes age, comorbidities, and gender. Are there other SES variables in NIS, such as SES, income, insurance type, that should be included in the adjustment? Agree that organ dysfunction should not be included in the regression model.

Minor revisions:

1. Title Line 1. Suggest changing patients to hospitalizations. The NIS includes hospitalizations in which one patient could be readmitted. Therefore, each hospitalization may not be unique to one patient.

2. Abstract Line 26. There are places (lines 27 and 37) where site of infection and source of infection are used interchangeably. Suggest using one consistent word such as site for all subsequent descriptions.

3. Introduction Line 49. “due to being one” sounds confusing. Suggest re-wording.

4. Intro Line 52. Suggest changing “Septicemia” to Sepsis.

5. Methods Line 78. Please provide rationale for study period 2006-2014. Is it because sepsis codes were developed in 2003 and there was late adoption? Also provide rationale for the break points chosen in Tables 1, 2, and 3.

6. Methods Line 94. Shouldn’t the acronym be EHR?

7. Methods Line 131. Consider making significance level <0.01 given size of sample (see PMID:30398593).

8. Results line 143-144. Is death number the weighted estimate of in-hospital mortality or the raw number from NIS? Would suggest rewording death number.

9. Results Line 163. Change intra-abdomen to intra-abdominal

10. Discussion Lines175-176. Reword septic source

11. Discussion Line 177. Which studies? Please include references

12. Discussion Lines 193-194. I’m not sure the urinary system has more protective barriers than the respiratory system. This sentence sounds strange. Would re-word.

13. Discussion Line 193. Remove period after mortality

14. Discussion Line 203. Suggest re-writing. What does tolerate mean?

15. Discussion Line 211. Change HER to EHR

16. Conclusion Lines 220-225. This feels vague. What specific trend do you want readers to take away and what specific future advancement does this study help with?

17. Tables 1, 2, and 3. Are these all weighted numbers? Please clarify. Suggest changing to title to reflect weighted analysis

18. Table 1. What comorbidity score was chosen? Please reference in methods.

19. Table 2. Change “systematic” to systemic. Double check throughout.

20. Figure 1. Change “miss” to missing values

21. Figure 2b. Y axis title is not fully seen

22. Figures. Suggest placing legends underneath graphs

23. Figure 4. I like this.

24. Supplementary Table 1. Why was primary bacteremia chosen as the reference? Primary bacteremia without a source seems like a rare entity and as such should not be the comparison.

25. Supplementary Table 1. Almost all variables reach statistical significance due to the sample size. Is there a comparison population you could track over the time period (i.e. those requiring mechanical ventilation) to confidently say the sepsis trend is changing rather than an artifact of the large sample size? This could be a “control group analysis” throughout the methods, results, and discussion.

6. PLOS authors have the option to publish the peer review history of their article (what does this mean?). If published, this will include your full peer review and any attached files.

Reviewer #1: Yes: Matthew K. Hensley

---

## [Author Response · Author response to Decision Letter 0]

12 Oct 2019

Ref: PONE-D-19-15923

Title: Incidence, trends, and outcomes of infection sites among hospitalizations of sepsis: a nationwide study

Journal: PLOS ONE

Dear editor and reviewers:

 Thank you for your e-mail with the referees’ comments. We have revised the manuscript as advised. The detailed point-by-point responses are as follows. We hope the manuscript is now acceptable to PLOS ONE. 

Editor’s Comment:

Thank you for submitting your manuscript to PLOS ONE. After careful consideration, we feel that it has merit but does not fully meet PLOS ONE’s publication criteria as it currently stands. Therefore, we invite you to submit a revised version of the manuscript that addresses the points raised during the review process.

The reviewers raised some concerns regarding the chosen sepsis definition, particularly as up to 30% of patients with sepsis do not have positive cultures and therefore would not be captured in the current analysis. Therefore, sensitivity analysis using different sepsis definition criteria (implicit Angus criteria vs. explicit sepsis criteria) should be performed to corroborate the presented results.

Author Reply:

Thanks for your comment. We have added our sensitivity analysis using Angus criteria into the supplementary section accordingly and added relevant description into our manuscript. Similar to our analysis, sensitivity analysis demonstrated increasing trend of sepsis. Meanwhile, the sensitivity test yielded similar result as our main analysis.

“Sensitivity analysis using Angus criteria was performed to corroborate the result.” (Section of Method, line 99)

 “Sensitivity analysis with Angus criteria showed similar trend of sepsis as our main results. (S4-S7 Table and S1-3 Figs)” (Section of Results, line 187-189)

In addition, ICD-9 codes used to define 'organ dysfunction' should be listed in the supplement. 

Author Reply:

Thanks for your advice. We have added into our manuscript as supporting table 2.

Second, it is unclear whether the analyses included both community-acquired sepsis ('present on admission') or nosocomial sepsis. These are very different entities with regards to epidemiology, source of infection, risk of death, etc. 

Author Reply:

Thanks for your advice. We agree that community-acquired and nosocomial sepsis have distinctive differences. However, due to the limitation of the database, we didn’t have sufficient data to determine whether it is community-acquired or nosocomial. We have added to our limitation section. 

“Meanwhile, due to the insufficient information from the database, we cannot identify patients’ socioeconomical status, community-acquired or nosocomial, medical or surgical hospitalization. Also, we didn’t perform the control group analysis due to insufficient data. Further study involving more detailed in-patient data is required.” (Section of Discussion, line 246-249)

Along these lines it would be interesting to know what proportion of hospitalizations were primary 'medical' or 'surgical'. 

Author Reply:

Thanks for your advice. Due to the limitation of the database, we don’t have sufficient data to determine whether it is surgical or medical. We have added to our limitation section. 

“Meanwhile, due to the insufficient information from the database, we cannot identify patients’ socioeconomical status, community-acquired or nosocomial, medical or surgical hospitalization. Also, we didn’t perform the control group analysis due to insufficient data. Further study involving more detailed in-patient data is required.” (Section of Discussion, line 246-249)

Third, previous studies by Lindenauer (JAMA 2012) and Rhee (JAMA 2017) suggest that temporal trends in pneumonia and sepsis estimates are associated with differences in coding - additional sensitivity analysis should be performed to verify or refute these findings.

Author Reply:

Thank you for your comment. We have done a sensitivity test using Angus criteria to verify our results. We have revised the manuscript and added the sensitivity test results in the supplementary section accordingly. 

“Sensitivity analysis using Angus criteria was performed to corroborate the result.” (Section of Method, line 99)

 “Sensitivity analysis with Angus criteria showed similar trend of sepsis as our main results. (S4-S7 Table and S1-3 Figs)” (Section of Results, line 187-189)

An additional limitation is the inability of administrative data to establish a firm temporal relationship between sepsis and the onset of organ dysfunction. This important limitation should be added to the discussion section. 

Author Reply:

Thanks for your advice. We have corrected the manuscript accordingly. 

“Moreover, our administrative data is unable to establish a firm temporal relationship between sepsis and the onset of organ dysfunction.” (Section of Discussion, line 245-246)

Finally, the discussion should be expanded to contrast the findings to preexisting literature. For example, Zahar JR et al previously reported no association between infection site and presence of bacteremia with mortality. 

Author Reply:

Thanks for your advice. We have revised the manuscript accordingly. 

“Zahar et al suggested that neither site of infection nor presence of bacteremia associated with mortality. (9) However, that study was a single center study with limited sample size. Our study used national database to expand sample size and increase statistic power.” (Section of Discussion, line 197-198) 

Journal Requirements:

2. Please include your tables as part of your main manuscript and remove the individual files. Please note that supplementary tables (should remain/ be uploaded) as separate "supporting information" files

This work is partly supported by NTUH.107-P03 grant

Author Reply:

Thanks for your advice. We didn’t receive any funding for this study. We have corrected the manuscript accordingly.

Author Reply:

Thanks for your advice. We didn’t receive any funding for this study. We have corrected the manuscript accordingly.

Reviewers' comments:

Reviewer's Responses to Questions

Comments to the Author

1. Is the manuscript technically sound, and do the data support the conclusions?

Reviewer #1: Partly

2. Has the statistical analysis been performed appropriately and rigorously? 

Reviewer #1: Yes

3. Have the authors made all data underlying the findings in their manuscript fully available?

Reviewer #1: Yes

4. Is the manuscript presented in an intelligible fashion and written in standard English?

Reviewer #1: Yes

5. Review Comments to the Author

Reviewer #1: Mann et al. have conducted a longitudinal analysis of sepsis outcomes by site of infection, over an extended period, using the National Inpatient Sample database, representing a large portion of the U.S. population. The authors have performed an appropriate analysis based on the data available and provided trends in mortality by infectious site over time, an important finding. However, I have some reservations about the methodology used provided in comments below.

Major revisions:

1. Methods: The authors’ primary conclusion in this study is that site of infection affects mortality and is changing over time. However, identification of site of infection seems inconsistent.

a. (Supplemental Table 2) Certain infections appear to be omitted from the inclusion diagnoses (i.e. cholangitis, Clostridium dificile colitis, CNS infections, endocarditis, and GU infections in women such as endometritis or ovarian abscess). Suggest improving identification of source of infection as this is the most important variable in the analysis.

Author Reply: 

Thank you for your comment. We used Martin’s criteria for the identification of sepsis. Because Martin’s criteria did not provide detailed definitions of different infection sites, we identified infection sites based on Angus ICD9-CM Sepsis Abstraction Criteria. (1) 

Angus et al performed a large-scale, multicenter epidemiological study and implemented the identification of severe sepsis using an ICD-9 based algorithm. (1) The Angus implementation is by far one of the most well-known and highly cited implementations of an ICD-coded case definition for sepsis. (2-6) In addition, using the same criteria as other studies could increase the comparability of our study, and would provide opportunity for future systemic review and meta-analysis. We have included in our limitation. 

“In order to increase the comparability of our study, we used Angus implementation for the recognition of infection sites. The Angus implementation was originally invented for the identification of severe sepsis with an ICD-9 based criteria, which is by far one of the most widely used implementations. (9, 34-38) Using the same criteria as other studies could increase the comparability and provide opportunity for future meta-analysis. However, some of the specific diagnoses may not be included.”

(Section of Discussion, line 250-255)

b. Second, how do the authors identify the primary site of infection? For example, if a hospitalization is coded for intestinal perforation and bacteremia and fungemia, how would this hospitalization be grouped in their analysis? Would it be the highest ranking diagnosis code or would it be included in all groups. This is important for assumptions of mortality in each group. It appears that these groups are mutually exclusive but it is not clear in the methods section.

Author Reply:

Thanks for your advice. Based on our method, we picked the top 2 (Primary and Secondary) diagnosis listed in the chart. The method has been clarified in the manuscript.

“For patient with multiple diagnoses, only primary and secondary diagnoses were recorded.” (Section of Method, line 109-110)

c. Does primary bacteremia mean no other diagnosis could be found for site of infection? More detail is required since there is a significant decrease in this site of infection over time. I suspect the outcomes and numbers of this site of infection would change greatly once item 1a is addressed.

Author Reply:

Thanks for your advice. We used the ICD-9 code 790.7 for the identification of primary bacteremia listed as primary diagnosis. No other diagnosis of infection sites was found.

2. Methods Line 95. The authors identify sepsis using methods by Martin et al. There are several other methods for identifying sepsis including Dombrovskiy et al. (PMID: 17414736). To improve the confidence of the conclusions, a sensitivity analysis using one of the other methods would be helpful.

Author Reply:

Thanks for your comment. We have added our sensitivity analysis using Angus criteria into the supplementary section accordingly and added relevant description into our manuscript. Similar to our analysis, sensitivity analysis demonstrated increasing trend of sepsis. Meanwhile, the sensitivity test yielded similar result as our main analysis.

“Sensitivity analysis using Angus criteria was performed to corroborate the result.” (Section of Method, line 99)

 “Sensitivity analysis with Angus criteria showed similar trend of sepsis as our main results. (S4-S7 Table and S1-3 Figs)” (Section of Results, line 187-189)

3. Methods Line 126. The adjusted analysis includes age, comorbidities, and gender. Are there other SES variables in NIS, such as SES, income, insurance type, that should be included in the adjustment? Agree that organ dysfunction should not be included in the regression model.

Author Reply:

Thanks for your advice. Due to the limitation of the database, we didn’t have sufficient data to determine socioeconomical status. We have added to our limitation section as follows. 

“Meanwhile, due to the insufficient information from the database, we cannot identify patients’ socioeconomical status, community-acquired or nosocomial, medical or surgical hospitalization. Also, we didn’t perform the control group analysis due to insufficient data. Further study involving more detailed in-patient data is required.” (Section of Discussion, line 246-249)

Minor revisions:

1. Title Line 1. Suggest changing patients to hospitalizations. The NIS includes hospitalizations in which one patient could be readmitted. Therefore, each hospitalization may not be unique to one patient.

Author Reply:

Thanks for your advice. We have corrected the manuscript accordingly. The title would be changed to “Incidence, Trends, and Outcomes of Infection Sites among Hospitalizations of Sepsis: a Nationwide Study.”

2. Abstract Line 26. There are places (lines 27 and 37) where site of infection and source of infection are used interchangeably. Suggest using one consistent word such as site for all subsequent descriptions.

Author Reply:

Thanks for your advice. We have corrected the manuscript accordingly. We have replaced all “source of infection” to “site of infection.”

3. Introduction Line 49. “due to being one” sounds confusing. Suggest re-wording.

Author Reply:

Thanks for your advice. We have corrected the manuscript accordingly. 

“Being one of the most expensive conditions to treat and a leading cause of death, sepsis has become a major health problem.” (Section of Introduction, line 53-54)

4. Intro Line 52. Suggest changing “Septicemia” to Sepsis.

Author Reply:

Thanks for your advice. We have corrected the manuscript accordingly. (Section of Introduction, line 56)

5. Methods Line 78. Please provide rationale for study period 2006-2014. Is it because sepsis codes were developed in 2003 and there was late adoption? Also provide rationale for the break points chosen in Tables 1, 2, and 3.

Author Reply:

Thanks for your advice. Because ICD-10 was initiated since 2015, we choose a 9-year study period between 2006 and 2014 in our study. In consideration of the readability, we divided our population into 3 groups with same period of time for further comparison.

6. Methods Line 94. Shouldn’t the acronym be EHR?

Author Reply:

Thanks for your advice. We have corrected the manuscript accordingly. 

7. Methods Line 131. Consider making significance level <0.01 given size of sample (see PMID:30398593).

Author Reply:

Thanks for your advice. We have corrected the manuscript accordingly. 

8. Results line 143-144. Is death number the weighted estimate of in-hospital mortality or the raw number from NIS? Would suggest rewording death number.

Author Reply:

Thanks for your advice. The death number is the weighted number. We have clarified in our manuscript.

“Taking the incidence and mortality rate together, lower respiratory tract infection was the leading cause of mortality (weighted death number=795,825), followed by genitourinary tract infection (weighted death number=489,964) and systemic fungal infection (weighted death number=153,027). (Section of Results, line 147-150)

9. Results Line 163. Change intra-abdomen to intra-abdominal

Author Reply:

Thanks for your advice. We have corrected the manuscript accordingly. 

10. Discussion Lines175-176. Reword septic source

Author Reply:

Thanks for your advice. We have corrected the manuscript accordingly. We have reworded it to “site of infection.”

11. Discussion Line 177. Which studies? Please include references

Author Reply:

Thanks for your advice. We have added the reference in the manuscript. 

12. Discussion Lines 193-194. I’m not sure the urinary system has more protective barriers than the respiratory system. This sentence sounds strange. Would re-word.

Author Reply:

Thanks for your advice. We have revised the manuscript accordingly as follows. 

“In contrast, those infections with multiple protective barriers (such as skin and musculoskeletal infections) had lower mortality rates (26).” (Section of Discussion, line 221-223)

13. Discussion Line 193. Remove period after mortality

Author Reply:

Thanks for your advice. We have corrected the manuscript accordingly.

14. Discussion Line 203. Suggest re-writing. What does tolerate mean?

Author Reply:

Thanks for your advice. We have re-written it.

“Those patient with lower risk of mortality might benefit from early de-escalation of the antibiotics.” (Section of Discussion, line 230-231)

15. Discussion Line 211. Change HER to HER

Author Reply:

Thanks for your advice. We have corrected the manuscript accordingly. 

16. Conclusion Lines 220-225. This feels vague. What specific trend do you want readers to take away and what specific future advancement does this study help with?

Author Reply:

Thank you for your comment. We have revised the manuscript.

“There is a significant difference in the trend of incidence and outcome of sepsis from different anatomic sites of infection. Clinician should be aware of different anatomic sites of infection could cause higher mortality in septic patients such as intra-abdominal infection, lower respiratory tract infection, and biliary tract infection” (Section of Conclusion, line 259-262)

“Conclusions: The anatomic site of infection does have a differential impact on the mortality of septic patients. Intra-abdominal infection, lower respiratory tract infection, and biliary tract infection are associated with higher mortality in septic patients.” (Section of Abstract, line 49-51)

17. Tables 1, 2, and 3. Are these all weighted numbers? Please clarify. Suggest changing to title to reflect weighted analysis

Author Reply:

Thanks for your advice. These are weighted numbers. We have revised the table title accordingly.

18. Table 1. What comorbidity score was chosen? Please reference in methods.

Author Reply:

Thanks for your advice. We have added the scoring index into the manuscript accordingly. 

“We use Elixhauser comorbidity Index as our comorbidity index.” (Section of Method, line 110-111)

19. Table 2. Change “systematic” to systemic. Double check throughout.

Author Reply:

Thanks for your advice. We have corrected the table accordingly. 

20. Figure 1. Change “miss” to missing values

Author Reply:

Thanks for your advice. We have corrected the figure accordingly. 

21. Figure 2b. Y axis title is not fully seen

Author Reply:

Thanks for your advice. We have corrected the figure accordingly. 

22. Figures. Suggest placing legends underneath graphs

Author Reply:

Thanks for your advice. We have corrected the figure accordingly. 

23. Figure 4. I like this.

Author Reply:

Thanks for your comment.

24. Supplementary Table 1. Why was primary bacteremia chosen as the reference? Primary bacteremia without a source seems like a rare entity and as such should not be the comparison.

Author Reply:

Thanks for your advice. Primary bacteremia was used as a reference because patient with only primary bacteremia had the lowest mortality. In order to make the chart more readable, we choose the lowest mortality one as the reference.

25. Supplementary Table 1. Almost all variables reach statistical significance due to the sample size. Is there a comparison population you could track over the time period (i.e. those requiring mechanical ventilation) to confidently say the sepsis trend is changing rather than an artifact of the large sample size? This could be a “control group analysis” throughout the methods, results, and discussion.

Author Reply:

Thank you for your comment. In the Supplementary Table 1, we focus on the mortality differences among different sites of infection. As for the trend of sepsis, there has been a debate in the trend of sepsis from prior literatures. (2-11) Rhee et al (12) found that the annul increase of sepsis was higher when using discharge code versus clinical criteria. So far Rhee et al (JAMA 2017) provided the best evidence that “neither the incidence of sepsis nor the combined outcome of death or discharge to hospice changed significantly.” (11) However, the primary aim of our study was to delineate the change in the incidence and in-hospital mortality between specific infection sites in sepsis over time, not the overall incidence of sepsis. Meanwhile, due to the limitation from our database, we are not able to provide such sophisticated results on this issue or control group analysis. We have listed it in the limitation section accordingly. Lastly, as we focused on the comparisons among different sites of infection, the results should not be affected significantly by the overall trend of sepsis or the definitive criteria applied. We have added this to our limitation as follows:

“Meanwhile, due to the insufficient information from the database, we cannot identify patients’ socioeconomical status, community-acquired or nosocomial, medical or surgical hospitalization. Also, we didn’t perform the control group analysis due to insufficient data. Further study involving more detailed in-patient data is required.” (Section of Discussion, line 246-249)

6. PLOS authors have the option to publish the peer review history of their article (what does this mean?). If published, this will include your full peer review and any attached files.

Do you want your identity to be public for this peer review? For information about this choice, including consent withdrawal, please see our Privacy Policy.

Reviewer #1: Yes: Matthew K. Hensley

1. Angus DC, Linde-Zwirble WT, Lidicker J, Clermont G, Carcillo J, Pinsky MR. Epidemiology of severe sepsis in the United States: analysis of incidence, outcome, and associated costs of care. Crit Care Med 2001; 29: 1303-1310.

2. Jolley RJ, Sawka KJ, Yergens DW, Quan H, Jette N, Doig CJ. Validity of administrative data in recording sepsis: a systematic review. Crit Care 2015; 19: 139.

3. Zahar JR, Timsit JF, Garrouste-Orgeas M, Francais A, Vesin A, Descorps-Declere A, Dubois Y, Souweine B, Haouache H, Goldgran-Toledano D, Allaouchiche B, Azoulay E, Adrie C. Outcomes in severe sepsis and patients with septic shock: pathogen species and infection sites are not associated with mortality. Crit Care Med 2011; 39: 1886-1895.

4. Stevenson EK, Rubenstein AR, Radin GT, Wiener RS, Walkey AJ. Two decades of mortality trends among patients with severe sepsis: a comparative meta-analysis*. Crit Care Med 2014; 42: 625-631.

5. Iwashyna TJ, Odden A, Rohde J, Bonham C, Kuhn L, Malani P, Chen L, Flanders S. Identifying patients with severe sepsis using administrative claims: patient-level validation of the angus implementation of the international consensus conference definition of severe sepsis. Med Care 2014; 52: e39-43.

6. Dombrovskiy VY, Martin AA, Sunderram J, Paz HL. Rapid increase in hospitalization and mortality rates for severe sepsis in the United States: a trend analysis from 1993 to 2003. Crit Care Med 2007; 35: 1244-1250.

7. Gaieski DF, Edwards JM, Kallan MJ, Carr BG. Benchmarking the incidence and mortality of severe sepsis in the United States. Crit Care Med 2013; 41: 1167-1174.

8. Lagu T, Rothberg MB, Shieh MS, Pekow PS, Steingrub JS, Lindenauer PK. Hospitalizations, costs, and outcomes of severe sepsis in the United States 2003 to 2007. Crit Care Med 2012; 40: 754-761.

9. Kumar G, Kumar N, Taneja A, Kaleekal T, Tarima S, McGinley E, Jimenez E, Mohan A, Khan RA, Whittle J, Jacobs E, Nanchal R. Nationwide trends of severe sepsis in the 21st century (2000-2007). Chest 2011; 140: 1223-1231.

10. Martin GS, Mannino DM, Eaton S, Moss M. The epidemiology of sepsis in the United States from 1979 through 2000. N Engl J Med 2003; 348: 1546-1554.

11. Rhee C, Dantes R, Epstein L, Murphy DJ, Seymour CW, Iwashyna TJ, Kadri SS, Angus DC, Danner RL, Fiore AE, Jernigan JA, Martin GS, Septimus E, Warren DK, Karcz A, Chan C, Menchaca JT, Wang R, Gruber S, Klompas M. Incidence and Trends of Sepsis in US Hospitals Using Clinical vs Claims Data, 2009-2014. Jama 2017; 318: 1241-1249.

12. Rhee C, Murphy MV, Li L, Platt R, Klompas M. Improving documentation and coding for acute organ dysfunction biases estimates of changing sepsis severity and burden: a retrospective study. Crit Care 2015; 19: 338.

---

## [Decision Letter · Decision Letter 1]

18 Nov 2019

PONE-D-19-15923R1

Incidence, Trends, and Outcomes of Infection Sites among Hospitalizations of Sepsis: a Nationwide Study

PLOS ONE

Dear Dr Lee,

Thank you for submitting your manuscript to PLOS ONE. After careful consideration, we feel that it has merit but does not fully meet PLOS ONE’s publication criteria as it currently stands. Therefore, we invite you to submit a revised version of the manuscript that addresses the points raised during the review process.

Please make sure to address the additional comments raised by the reviewers. In particular, please justify the use of odds / odds ratios for risk prediction  given its known limitations (e.g., Pepe MS, Am J Epidemiol 2004). Furthermore, please include sensitivity analyses using alternative sepsis coding strategy (e.g., 'Angus methodology') in the supplement section.

We would appreciate receiving your revised manuscript by Jan 02 2020 11:59PM. To enhance the reproducibility of your results, we recommend that if applicable you deposit your laboratory protocols in protocols.io, where a protocol can be assigned its own identifier (DOI) such that it can be cited independently in the future. For instructions see: http://journals.plos.org/plosone/s/submission-guidelines#loc-laboratory-protocols

We look forward to receiving your revised manuscript.

Kind regards,

Florian B. Mayr

Academic Editor

PLOS ONE

Reviewers' comments:

Reviewer's Responses to Questions

**Comments to the Author**

1. If the authors have adequately addressed your comments raised in a previous round of review and you feel that this manuscript is now acceptable for publication, you may indicate that here to bypass the “Comments to the Author” section, enter your conflict of interest statement in the “Confidential to Editor” section, and submit your "Accept" recommendation.

Reviewer #1: All comments have been addressed

2. Is the manuscript technically sound, and do the data support the conclusions?

Reviewer #1: Yes

3. Has the statistical analysis been performed appropriately and rigorously? 

Reviewer #1: Yes

4. Have the authors made all data underlying the findings in their manuscript fully available?

Reviewer #1: Yes

5. Is the manuscript presented in an intelligible fashion and written in standard English?

Reviewer #1: Yes

6. Review Comments to the Author

Reviewer #1: Chou et al. have significantly improved the methodology of this study. The interpretation of results is sound with expanded limitations and discussion. I agree with publication of these important results with minor revisions listed below.

Minor revisions:

1) Line 151-152: This sentence is confusing. Please re-word. “Male patients tend to be more likely to sepsis in all subperiods.”

2) Lines 182-189: You use Odds Ratios, but in the methods section lines 133-134 you mention that odds ratios are biased measures given the prevalence of sepsis and therefore do not reliably predict risk. Please clarify.

3) Lines 209-210: “One example being the surviving sepsis campaign being started at the beginning of this study who could account for increased incidence of sepsis”. This is confusing. Please re-word.

4) Lines 214-215: “which is common etiologic agent of pneumonia.” Please fix grammar.

7. PLOS authors have the option to publish the peer review history of their article (what does this mean?). If published, this will include your full peer review and any attached files.

Reviewer #1: Yes: Matthew K Hensley

---

## [Author Response · Author response to Decision Letter 1]

2 Dec 2019

Ref: PONE-D-19-15923

Title: Incidence, trends, and outcomes of infection sites among hospitalizations of sepsis: a nationwide study

Journal: PLOS ONE

Dear editor and reviewers:

 Thank you for your e-mail with the referees’ comments. We have revised the manuscript as advised. The detailed point-by-point responses are as follows. We hope the manuscript is now acceptable to PLOS ONE. 

Editor’s Comment:

Thank you for submitting your manuscript to PLOS ONE. After careful consideration, we feel that it has merit but does not fully meet PLOS ONE’s publication criteria as it currently stands. Therefore, we invite you to submit a revised version of the manuscript that addresses the points raised during the review process.

Please make sure to address the additional comments raised by the reviewers. In particular, please justify the use of odds / odds ratios for risk prediction given its known limitations (e.g., Pepe MS, Am J Epidemiol 2004). Furthermore, please include sensitivity analyses using alternative sepsis coding strategy (e.g., 'Angus methodology') in the supplement section.

Author Reply:

Thanks for your comment. To evaluate the impact of individual site of infection on the survival of sepsis patients, we fit a multivariable logistic regression model adjusting for age, sex, and comorbidity measures. We used the entire study period for this regression analysis to ensure adequate power to make reliable estimates of risk. Because the mortality rate for patients with sepsis is higher than 10% in this analysis, the rare disease assumption does not hold. As a result, risk ratios cannot be estimated by odds ratios. Therefore, we used the formula proposed by Zhang and Yu to approximate the relative risk. (1) We have corrected our manuscript and figures accordingly.

“As a result, risk ratios cannot be estimated by odds ratios.” (Line 134 in Section of Methods) 

“Figure 4 shows the adjusted relative risk with 95% confidence intervals of infection site on the outcome of sepsis. Using primary bacteremia as reference, sepsis patients with intra-abdominal infection had the highest mortality (RR:4.21), followed by lower respiratory tract infection (RR: 3.84), biliary tract infection (RR: 3.24), systemic fungal infection (RR: 2.77), skin and skin structure infection (RR: 2.29), musculoskeletal infection (RR: 2.27), genitourinary traction infection (RR:2.19), or catheter related bloodstream infection (RR:2.15). Sensitivity analysis with Angus criteria showed similar trend of sepsis as our main results (Table S4-7 and Fig S1-3)

Fig 4. Survival impact of individual infection site in relation to primary bacteremia. The risk estimates were adjusted for all covariates listed in supporting table 1. RR refers to the relative risk. LCL and UCL refer to lower and upper confidence limits, respectively.” (Line 182-191 in Section of Results)

“Supporting Figure 3. Sensitivity Test - Survival impact of individual infection site in relation to primary bacteremia. The risk estimates were adjusted for all covariates listed in Supporting table 7. RR refers to the relative risk. LCL and UCL refer to lower and upper confidence limits, respectively.

Supporting Table 1. Covariates with associated relative risk in the outcome regression model.” (Line 371-376 in Section of Supporting Information)

“Supporting Table 7. Sensitivity Test - Covariates with associated relative risk in the outcome regression model.” (Line 386-388 in Section of Supporting Information)

We have included sensitivity analyses using Angus methodology in Table S4-7 and Fig S1-3.

We would appreciate receiving your revised manuscript by Jan 02 2020 11:59PM. To enhance the reproducibility of your results, we recommend that if applicable you deposit your laboratory protocols in protocols.io, where a protocol can be assigned its own identifier (DOI) such that it can be cited independently in the future. For instructions see: http://journals.plos.org/plosone/s/submission-guidelines#loc-laboratory-protocols

• A rebuttal letter that responds to each point raised by the academic editor and reviewer(s). This letter should be uploaded as separate file and labeled 'Response to Reviewers'.

• A marked-up copy of your manuscript that highlights changes made to the original version. This file should be uploaded as separate file and labeled 'Revised Manuscript with Track Changes'.

• An unmarked version of your revised paper without tracked changes. This file should be uploaded as separate file and labeled 'Manuscript'.

We look forward to receiving your revised manuscript.

Kind regards,

Florian B. Mayr

Academic Editor

PLOS ONE

Reviewers' comments:

Reviewer's Responses to Questions

Comments to the Author

1. If the authors have adequately addressed your comments raised in a previous round of review and you feel that this manuscript is now acceptable for publication, you may indicate that here to bypass the “Comments to the Author” section, enter your conflict of interest statement in the “Confidential to Editor” section, and submit your "Accept" recommendation.

Reviewer #1: All comments have been addressed

2. Is the manuscript technically sound, and do the data support the conclusions?

Reviewer #1: Yes

3. Has the statistical analysis been performed appropriately and rigorously? 

Reviewer #1: Yes

4. Have the authors made all data underlying the findings in their manuscript fully available?

Reviewer #1: Yes

5. Is the manuscript presented in an intelligible fashion and written in standard English?

Reviewer #1: Yes

6. Review Comments to the Author

Reviewer #1: Chou et al. have significantly improved the methodology of this study. The interpretation of results is sound with expanded limitations and discussion. I agree with publication of these important results with minor revisions listed below.

Minor revisions:

1) Line 151-152: This sentence is confusing. Please re-word. “Male patients tend to be more likely to sepsis in all subperiods.”

Author Reply:

Thanks for your comment. We have re-worded accordingly. “There are more male patients than female in all subperiods.” (Line 151-152 in Section of Results)

2) Lines 182-189: You use Odds Ratios, but in the methods section lines 133-134 you mention that odds ratios are biased measures given the prevalence of sepsis and therefore do not reliably predict risk. Please clarify.

Author Reply:

Thanks for your comment. To evaluate the impact of individual site of infection on the survival of sepsis patients, we fit a multivariable logistic regression model adjusting for age, sex, and comorbidity measures. We used the entire study period for this regression analysis to ensure adequate power to make reliable estimates of risk. Because the mortality rate for patients with sepsis is higher than 10% in this analysis, the rare disease assumption does not hold. As a result, risk ratios cannot be estimated by odds ratios. Therefore, we used the formula proposed by Zhang and Yu to approximate the relative risk. (1) We have corrected our manuscript, tables and figures accordingly.

 “As a result, risk ratios cannot be estimated by odds ratios.” (Line 134 in Section of Methods) 

“Figure 4 shows the adjusted relative risk with 95% confidence intervals of infection site on the outcome of sepsis. Using primary bacteremia as reference, sepsis patients with intra-abdominal infection had the highest mortality (RR:4.21), followed by lower respiratory tract infection (RR: 3.84), biliary tract infection (RR: 3.24), systemic fungal infection (RR: 2.77), skin and skin structure infection (RR: 2.29), musculoskeletal infection (RR: 2.27), genitourinary traction infection (RR:2.19), or catheter related bloodstream infection (RR:2.15). Sensitivity analysis with Angus criteria showed similar trend of sepsis as our main results (Table S4-7 and Fig S1-3)

Fig 4. Survival impact of individual infection site in relation to primary bacteremia. The risk estimates were adjusted for all covariates listed in supporting table 1. RR refers to the relative risk. LCL and UCL refer to lower and upper confidence limits, respectively.” (Line 182-191 in Section of Results)

“Supporting Figure 3. Sensitivity Test - Survival impact of individual infection site in relation to primary bacteremia. The risk estimates were adjusted for all covariates listed in Supporting table 7. RR refers to the relative risk. LCL and UCL refer to lower and upper confidence limits, respectively.

Supporting Table 1. Covariates with associated relative risk in the outcome regression model.” (Line 371-376 in Section of Supporting Information)

“Supporting Table 7. Sensitivity Test - Covariates with associated relative risk in the outcome regression model.” (Line 386-388 in Section of Supporting Information)

3) Lines 209-210: “One example being the surviving sepsis campaign being started at the beginning of this study who could account for increased incidence of sepsis”. This is confusing. Please re-word.

Author Reply:

Thanks for your comment. We have re-worded accordingly. 

“For example, the surviving sepsis campaign started at the beginning of this study could account for early sepsis recognition and decreasing mortality with early antibiotic administration and three-hour bundle therapy (20-22).” (Line 208-210 in Section of Discussions)

4) Lines 214-215: “which is common etiologic agent of pneumonia.” Please fix grammar.

Author Reply:

Thanks for your comment. We have re-worded accordingly. 

“Currently preventative strategies mostly aimed at streptococcus pneumoniae, which is the common pathogen of pneumonia.” (Line 212-213 in Section of Discussions)

7. PLOS authors have the option to publish the peer review history of their article (what does this mean?). If published, this will include your full peer review and any attached files.

Do you want your identity to be public for this peer review? For information about this choice, including consent withdrawal, please see our Privacy Policy.

Reviewer #1: Yes: Matthew K Hensley

Reference:

1. Zhang J, Yu KF. What's the relative risk? A method of correcting the odds ratio in cohort studies of common outcomes. Jama 1998; 280: 1690-1691.

---

## [Editor Report · Decision Letter 2]

30 Dec 2019

Incidence, Trends, and Outcomes of Infection Sites among Hospitalizations of Sepsis: a Nationwide Study

PONE-D-19-15923R2

Dear Dr. Lee,

We are pleased to inform you that your manuscript has been judged scientifically suitable for publication and will be formally accepted for publication once it complies with all outstanding technical requirements.

With kind regards,

Florian B. Mayr

Academic Editor

PLOS ONE
---

## [Editor Report · Acceptance letter]

6 Jan 2020

PONE-D-19-15923R2 

Incidence, Trends, and Outcomes of Infection Sites among Hospitalizations of Sepsis: a Nationwide Study 

Dear Dr. Lee:

I am pleased to inform you that your manuscript has been deemed suitable for publication in PLOS ONE. Congratulations! Your manuscript is now with our production department. 

With kind regards,

on behalf of

Dr. Florian B. Mayr 

Academic Editor

PLOS ONE